# Correlating IgG Levels with Neutralising Antibody Levels to Indicate Clinical Protection in Healthcare Workers at Risk during a Measles Outbreak

**DOI:** 10.3390/v14081716

**Published:** 2022-08-04

**Authors:** Siyuan Hu, Nicola Logan, Sarah Coleman, Cariad Evans, Brian J. Willett, Margaret J. Hosie

**Affiliations:** 1MRC-University of Glasgow Centre for Virus Research, Bearsden Road, Glasgow G61 1QH, UK; 2Virology Department, Sheffield Teaching Hospitals NHS Foundation Trust, Northern General Hospital, Herries Road, Sheffield S5 7AU, UK

**Keywords:** measles outbreak, healthcare workers, neutralising antibody, clinical protection, IgG testing

## Abstract

The rapid transmission of measles poses a great challenge for measles elimination. Thus, rapid testing is required to screen the health status in the population during measles outbreaks. A pseudotype-based virus neutralisation assay was used to measure neutralising antibody titres in serum samples collected from healthcare workers in Sheffield during the measles outbreak in 2016. Vesicular stomatitis virus (VSV) pseudotypes bearing the haemagglutinin and fusion glycoproteins of measles virus (MeV) and carrying a luciferase marker gene were prepared; the neutralising antibody titre was defined as the dilution resulting in 90% reduction in luciferase activity. Spearman’s correlation coefficients between IgG titres and neutralising antibody levels ranged from 0.40 to 0.55 (*p* < 0.05) or from 0.71 to 0.79 (*p* < 0.0001) when the IgG titres were obtained using different testing kits. In addition, the currently used vaccine was observed to cross-neutralise most circulating MeV genotypes. However, the percentage of individuals being “well-protected” was lower than 95%, the target rate of vaccination coverage to eliminate measles. These results demonstrate that the level of clinical protection against measles in individuals could be inferred by IgG titre, as long as a precise correlation has been established between IgG testing and neutralisation assay; moreover, maintaining a high vaccination coverage rate is still necessary for measles elimination.

## 1. Introduction

Measles is perhaps the most contagious infectious disease, with an often-cited R_0_ of 12–18 [1,2]. Exiting through epithelial cells of the trachea is a highly efficient way of spreading aerosol-transmitted viruses, such as measles virus (MeV). MeV binding to nectin-4, the epithelial receptor, and utilising it as a site-directed host exit receptor is thought to be advantageous, accounting for the very high transmission rate [3,4,5]. The most contagious period of measles is thought to start four days before the rash develops and to extend until four days after the rash appears; the finding that aerosol droplets could hang in the air for up to two hours means that susceptible individuals can be infected through indirect contact [6]. Fortunately, measles is vaccine preventable. Since measles vaccination was introduced in 1968, it has been estimated that 20 million cases and 4500 deaths have been prevented in the UK [7]. The introduction of the live-attenuated measles-mumps-rubella (MMR) trivalent vaccine in 1988, followed by the two-dose regime in 1996, led to a marked reduction in measles transmission, with national vaccination coverage rates exceeding 90%. It was observed that approximately 85% of vaccinated children produce a protective antibody response after one dose, and so the second dose was designed to immunise the remaining 15% [6].

In recent years, measles has re-emerged, largely due to inadequate vaccine coverage. Since 2016, measles outbreaks have been reported in multiple countries in Europe, such as Romania, Italy and France [8,9,10]. In 2020, over 12,000 cases were reported in 71% member states that submitted case data in the WHO European Region, of note, 61% of cases were reported in children less than five years old and 85% of patients were known to have been vaccinated [11]. During the COVID-19 pandemic, the number of measles cases declined in 2021, with only 70 cases reported to the European Region for the first half of the year [12].

Outbreaks in the UK have been associated with imported measles cases and have been predominantly caused by the B3 genotype associated with outbreaks throughout Europe [13]. Outbreaks involving the D8 genotype, commonly found in Southeast Asia, have also been reported [14]. During the summer of 2016, an outbreak of the D8 genotype occurred within Sheffield Teaching Hospitals, an acute healthcare trust in South Yorkshire. This outbreak involved over 4000 patient contacts and nearly 900 staff contacts from a single index case, culminating in 17 laboratory-confirmed cases of measles. Healthcare workers (HCWs) represent a special group at risk of intense exposure during measles outbreaks, with less opportunity for avoidance compared to people with other occupations. A study of measles outbreaks within healthcare settings in Washington, conducted in 1996, estimated that measles was 18.6 times more likely to occur in HCWs than the local adult population [15]. During this outbreak in Sheffield, several HCWs developed measles, despite having received two doses of measles vaccine. More seriously, several studies have demonstrated onward MeV transmission from vaccinated individuals [16,17,18]. Hence, rapid screening for antibody levels in the healthy population, such as HCWs, is important for quickly assessing susceptible individuals when a measles outbreak occurs.

The enzyme-linked immunosorbent assay (ELISA) is widely used as a clinical routine test because it is rapid and has been well standardised. However, the ELISA mainly measures antibodies against native virus antigens, and predominantly detects antibodies to the viral nucleoprotein, rather than the haemagglutinin (H) and fusion (F) glycoproteins that relate to clinical protection [19,20]. The neutralisation test is regarded as the gold standard test because it measures functional neutralising antibodies that bind directly to the surface glycoproteins H and F that are responsible for receptor attachment and host cell entry of MeV [21,22]. However, the neutralisation test is not suitable for clinical use because it has not been internationally standardised and has a turnaround time of several days [23]. In this case, a strong correlation between ELISA results and neutralising antibody titres would make it possible to assess the clinical immune response using the existing routine ELISA test for measles IgG.

In this study, we measured neutralising antibody responses against MeV in serum samples collected from HCWs in Sheffield during the measles outbreak. Neutralising antibody titres were determined using a pseudotype-based virus neutralisation assay (PVNA). Neutralising antibody titres were assessed against the Edmonston strain (vaccine strain) as well as representative field strains of genotypes B3, C2, D4, D8 and H1. B3 and D8 were reported to cause over 99% of measles cases in 2019, while the remaining cases were caused by D4 and H1; C2 is reported to have been eliminated [24]. The aim of this study was to compare total measles IgG titres with the MeV specific neutralising antibody titres and to determine whether it is possible to estimate the level of clinical protection based on the IgG titre in HCWs who have been vaccinated against measles.

## 2. Materials and Methods

### 2.1. Serum Samples

Serum samples from Sheffield were collected from staff members who had been in contact with confirmed measles cases and who either had no vaccination history or had only received one dose of MMR vaccine. These serum samples were tested for IgG titres to confirm the measles immune status of the HCWs during the outbreak that occurred from July to September 2016.

The protective threshold above which neutralising antibody levels protect against measles infection used in this study was 120 mIU/mL [25]. The neutralising antibody titre of the protection threshold was measured in the WHO 3rd International Standard Biological Reference human anti-measles serum (NIBSC 97/648), diluted to 120 mIU/mL.

### 2.2. Cell Lines

HEK 293T [26] and HEK 293 [27] that had been engineered to express human signalling lymphocyte activation molecule 1, so called SLAMF1, were maintained in Dulbecco’s Modified Eagle Medium supplemented with 10% foetal bovine serum, 100 IU/mL penicillin–streptomycin and 2 mM glutamine. Media for HEK 293-SLAMF1 cells were supplemented with 1 μg/mL puromycin and media for HEK 293T cells were supplemented with 400 μg/mL G418. All media and supplements were obtained from Thermo Fisher Scientific Ltd., Paisley, UK.

### 2.3. IgG Testing

The levels of IgG antibody in the serum samples from Sheffield were measured using LIAISON^®^ Measles IgG (Code 318810) (DiaSorin Ltd., Dartford, UK) and EUROIMMUN^®^ Anti-Measles Virus ELISA (EI 2610-9601 G) (EUROIMMUN UK Ltd., London, UK).

### 2.4. Production of Vesicular Stomatitis Virus (Measles Virus) Pseudotypes (VSV (MeV))

VSV (MeV) pseudotypes were prepared from six strains of MeV, namely Edmonston (contained in MMR vaccine), B3 (MVi/Rotterdam.NLD/32.12/1), C2 (MVi/Bilthoven.NLD/20.11), D4 (MVi/Amsterdam.NLD/19.11), D8 (MVi/Dodewaard.NLD/29.13) and H1 (MVi/Amsterdam.NLD/27.97), kindly provided by Dr. Rik de Swart, EMC, Rotterdam. The H genes were amplified and used to prepare VSV (MeV) pseudotypes using the method described previously [28].

### 2.5. Pseudotype-Based Virus Neutralisation Assay

A dilution series was prepared ranging from 1:32, then as four-fold from 1:64 to 1:65,536 for each serum sample, in triplicate. Detailed steps of the assay were described previously [29,30]. The luciferase activity was measured using a Chameleon^TM^ luminometer (LabLogic Systems Ltd., Sheffield, UK). Neutralising antibody titres were calculated by interpolating the point at which there was a 90% reduction in luciferase activity (90% neutralisation, inhibitory concentration 90 or IC90). The VSV (morbillivirus) pseudotype-based virus neutralisation assay was compared previously to live virus neutralisation assay and a good correlation was observed, indicating that the pseudotype assay is a reliable substitute for live virus-based assay [28].

### 2.6. Statistical Analysis

The neutralising antibody titre of each sample against different genotypes, which were non-normally distributed, was statistically compared with the titre against the vaccine strain, using the Wilcoxon matched-pairs signed-rank test to determine any significant difference [31].

Spearman’s rank correlation coefficients (R_s_) were calculated between IgG level and neutralising antibody titres [32].

## 3. Results

### 3.1. Immune Status Check in Sheffield HCWs during the Sheffield Measles Outbreak

The neutralisation activity against six MeV strains, namely Edmonston, B3, C2, D4, D8 and H1, were measured in serum samples collected from 64 HCWs whose immune status was checked and five patients who were diagnosed during the measles outbreak and titres were compared against the 3rd WHO reference serum (Figure 1a). A range of neutralising antibody titres was observed amongst individuals, with some individuals displaying titres below that of the 3rd WHO reference serum. The Edmonston titres were significantly lower than the titres against the other five pseudotypes tested (*p* < 0.0001 against B3, C2, D8 and H1, *p* = 0.0010 against D4).

The percentage of individuals with neutralising antibody titres greater than or equal to the protective threshold was calculated for each genotype tested; 87.3% of HCWs tested showed protective titres against Edmonston, 93.6% against B3, 87.3% against C2, 82.5% against D4, 88.9% against D8 and 88.9% against H1 (Table 1).

Figure 1b shows the results of both HCWs and five measles patients (A to E) who had been vaccinated previously. Patients A–D had been vaccinated with two doses of MMR vaccine, whereas patient E had received only one dose. These five patients displayed relatively strong neutralising antibody responses against the strains tested; patient E displayed the lowest titres compared to the other four patients.

### 3.2. Correlation between Measles-Specific IgG Level and Neutralising Antibody Titre: Relationship with Clinical Protection Status

The sera from the Sheffield HCWs were screened using a recombinant nucleoprotein-based measles-specific IgG immunoassay (LIAISON^®^ Measles IgG) at the time of the outbreak. Due to the limitations of the assay kit, precise values were obtained only when the IgG level was below 300 AU/mL. Figure 2 shows the results from 37 of 64 HCWs that were within the measurable range. The correlation between IgG levels and neutralising antibody titres measured by PVNA was analysed using Spearman’s rank correlation coefficient (R_s_) (Table 2). Samples with low IgG titres tended to show poor neutralising activity, while the neutralising antibody titres of samples with high IgG titres were highly variable against all pseudotypes tested (Figure 2).

A second ELISA kit, from EUROIMMUN^®^, was used to evaluate the samples from the Sheffield cohort as precise values could be obtained for IgG levels greater than 50 IU/L and less than 5000 IU/L. Figure 3 shows data from 61 of 68 samples collected from HCWs and Spearman’s rank correlation coefficients are shown in Table 3 and present a stronger positive correlation than what was observed with values obtained using the LIAISON^®^ kit.

## 4. Discussion

### 4.1. Vaccinated Individuals Could Still Be Infected with Measles

The immune status of HCWs was checked during the measles outbreak in Sheffield as they were defined as contacts. In the assessment of HCW contacts, satisfactory evidence of protection includes documentation of having received two or more doses of measles-containing vaccine and/or a positive measles IgG antibody. If HCWs are tested rapidly post exposure and are measles IgG positive, they can continue to work; if negative, they require assessment for post exposure prophylaxis with MMR vaccine and exclusion from work [33].

The HCWs had been routinely vaccinated with MMR and cross-neutralising antibodies were detected against most MeV genotypes that have been circulating worldwide. B3 was a widely circulating genotype in the UK for several years, and was isolated during recent outbreaks [13,34,35], while the D8 genotype caused the Sheffield outbreak in 2016. We observed that the titres against five circulating strains were significantly higher than the titre against the Edmonston (vaccine strain) pseudotype, with the percentage of individuals being protected against B3 being higher than the percentages against the other five strains. Given that B3 has been the dominant circulating genotype in the UK in recent years, re-exposure to B3 could be one of the explanations for the stronger protection in individuals against this genotype. Moreover, a notable number of patients from the Sheffield outbreak who were recorded as vaccinated displayed milder symptoms and were less infectious following infection than unvaccinated patients, which is consistent with previous reports of a lack of onward transmission reported in breakthrough cases [36,37,38,39]. It is tempting to speculate that re-exposure of vaccinated individuals might not lead to further transmission, but rather elicits stronger protection.

The required level of immunisation coverage to achieve measles elimination is defined as 95% [40], assuming that all vaccinated individuals develop protective immunity. The proportion of the samples tested with neutralising antibody titres within the “well-protected” range ranged from 82.5% to 93.6% for different genotypes, which fell below the 95% threshold required to eliminate measles. This finding suggests that, although cross-neutralisation was observed between the different MeV genotypes tested, a vaccination rate greater than 95% will be required to achieve protective immunity against MeV infection in at least 95% of the population.

The individuals with neutralising antibody titres below the protective threshold should be studied further. It is possible that those individuals had not been exposed to MeV, which would have boosted the antibody response following vaccination; it is likely that the natural waning of vaccine-induced immunity over time in the absence of exposure was the main reason that vaccinated individuals developed measles during the outbreak.

As the patient samples from the Sheffield outbreak were collected after the measles cases had been identified, the neutralising antibody titres of the measles patients who have previously been vaccinated were greater than the protective threshold. It is likely that these high titres had been induced following natural exposure during the measles outbreak rather than prior vaccination. This finding suggests that these patients should have the potential to develop higher levels of immunity in response to another dose of MMR vaccine when their immunity has waned following previous vaccination in the absence of re-exposure.

There are two factors that may cause persistent outbreaks within vaccinated populations. One is the inadequate vaccination coverage and the waning of the vaccine-induced immunity, which has been discussed above; the other one is the mutations within immunodominant sites of the glycoproteins, which is one of the most concerning problems during measles elimination. However, reported in 2019, there were only four genotypes still detected worldwide while 20 out of 24 recognised genotypes were thought to be eliminated since 2005 due to the immunisation [24]. Most cases related to B3 (22%) and D8 (78%) while a few were caused by D4 (0.1%) and H1 (0.3%). The elimination of most genotypes is consistent with what was concluded in this study that the currently used vaccine is efficient to cross-neutralise against most circulating genotypes in the last decades. However, the circulating strains used in the experiment were isolated no later than 2012 and have been replaced by more recent variants [41]. Thus, further assays to cross-neutralise against latest variants are necessary for further evaluation of the immunogenicity of the vaccine.

### 4.2. Correlation between IgG Levels and Neutralising Antibody Titres in Healthy Individuals Might Be Used to Predict Clinical Protection

Commercial IgG test kits are used widely to obtain results rapidly. However, total measles IgG titres do not indicate clinical protection status, which is generally acknowledged to be defined by neutralising antibody activity. The different R_s_ values observed between IgG titres and neutralising antibody titres could be attributed to the different sensitivities of different IgG assays and the fact that functional neutralising antibody represents only a subset of the total measles IgG. The sensitivity of the enzyme immunoassay for IgG was approximately 90% and sera containing low neutralising antibody levels have been shown to test negative for IgG [42,43]. In contrast, it is generally acknowledged that the neutralisation assay is one of the most sensitive and accurate measures of protective antibody levels [25]. However, neutralisation assays are more labour intensive and costly than enzyme immunoassays, which are largely automated. Given the rapid transmissibility of MeV, neutralisation assays are not commonly used in diagnostic laboratories because of the longer turnaround time.

Therefore, rapid IgG tests could prove useful to indicate clinical protection if IgG titres correlated strongly with neutralising antibody titres. Neutralising antibody titres of samples with IgG levels, which were measured by the kit from LIAISON^®^), within the measurable range (≤300 AU/mL) were plotted and analysed by Spearman’s rank correlation coefficient, the R_s_ values ranged from 0.4 to 0.55 against the tested pseudotypes; all *p* values indicated the correlation was significant (*p* < 0.05).

Another ELISA kit, from EUROIMMUN^®^, was used to re-test the IgG level in the Sheffield samples and a stronger correlation coefficient was observed. The R_s_ values ranged from 0.71 to 0.79 with *p* values indicating stronger significance (*p* < 0.0001), compared to the results obtained using LIAISON^®^ assay. Although the units IU/L and AU/mL are not comparable, more precise values being obtained using EUROIMMUN^®^ assay, indicating that the measuring range of the EUROIMMUN^®^ assay (50–5000 IU/L) is wider than that of the LIAISON^®^ assay (5–300 AU/mL). Moreover, several individuals with high IgG levels measured using LIAISON^®^ assay displayed neutralising antibody titres that were lower than the protective threshold, which was rarely observed in the results obtained using the EUROIMMUN^®^ assay, indicating that the EUROIMMUN^®^ assay might be more specific than the LIAISON^®^ assay. Hence, it is feasible to measure IgG levels to predict the clinical protection status, providing the results of IgG testing correlate well with neutralising antibody levels as we found using the EUROIMMUN^®^ assay.

An alternative method for health status screening could be to develop novel lateral flow tests with specific cut-off values to detect measles-specific IgG. These tests could effectively identify the target population requiring further immunisation and so inform vaccination campaigns. The cut-off values would be established based on a general protective threshold defined by WHO or on regional distributions of antibody titres among individuals. As measles still occurs in many countries around the world, the immune status of different ages within the population will be strongly affected by historical outbreaks and previous vaccination protocols. Therefore, the model set for establishing the appropriate cut-off point should be tailored to the local immunisation strategies and include representative samples from appropriate age groups.

## 5. Conclusions

The results of this study demonstrate the possibility of correlating IgG level with neutralising antibody titre; however, the accuracy of IgG level indicating protection is highly dependent on the characteristics of the IgG test kit used. Therefore, the level of clinical protection against measles in individuals could be inferred by IgG measuring, as long as a precise correlation has been established between the IgG measuring and neutralisation activity.

## Figures and Tables

**Figure 1 viruses-14-01716-f001:**
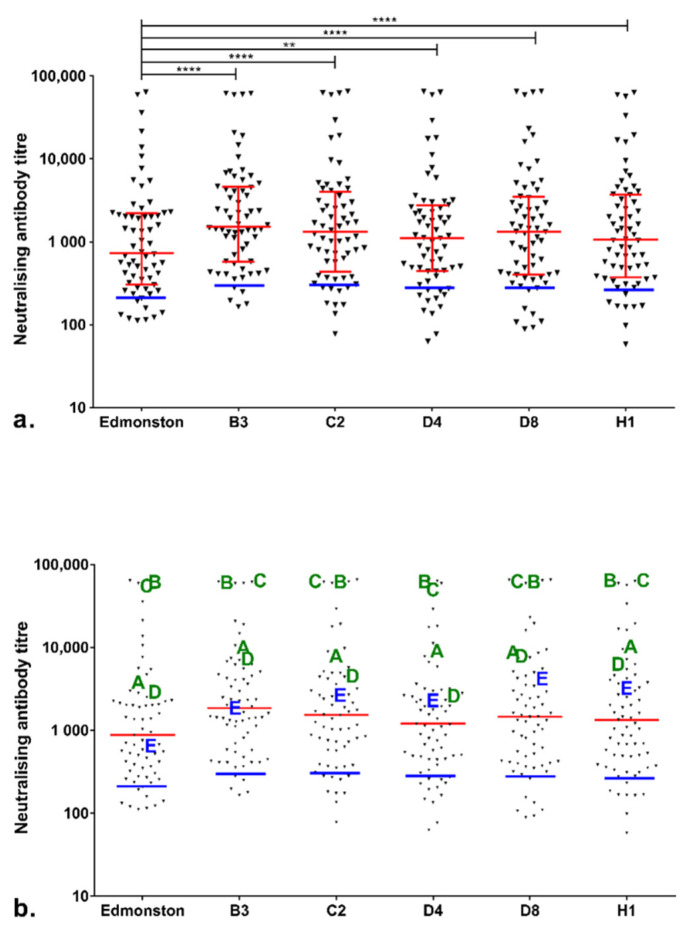
(**a**) Cross-neutralising antibody titres against MeV genotypes in 64 Sheffield HCWs who had their measles immune status checked at the time of the outbreak. Median titres with interquartile ranges are shown in red and Wilcoxon matched-pairs signed-rank test was used for statistical analysis. The blue line in each column indicates the titres of the 3rd WHO measles antibody international standard serum (NIBSC 97/648) diluted to 120 mIU/mL, defined as the protective threshold. Asterisks indicate statistical significance (**** *p* < 0.0001; ** *p* < 0.01). (**b**) Neutralising antibody titres of 64 HCWs and 5 patients displaying measles symptoms. Patients A–D had been vaccinated with two doses of MMR vaccine, whereas patient E had received only one dose.

**Figure 2 viruses-14-01716-f002:**
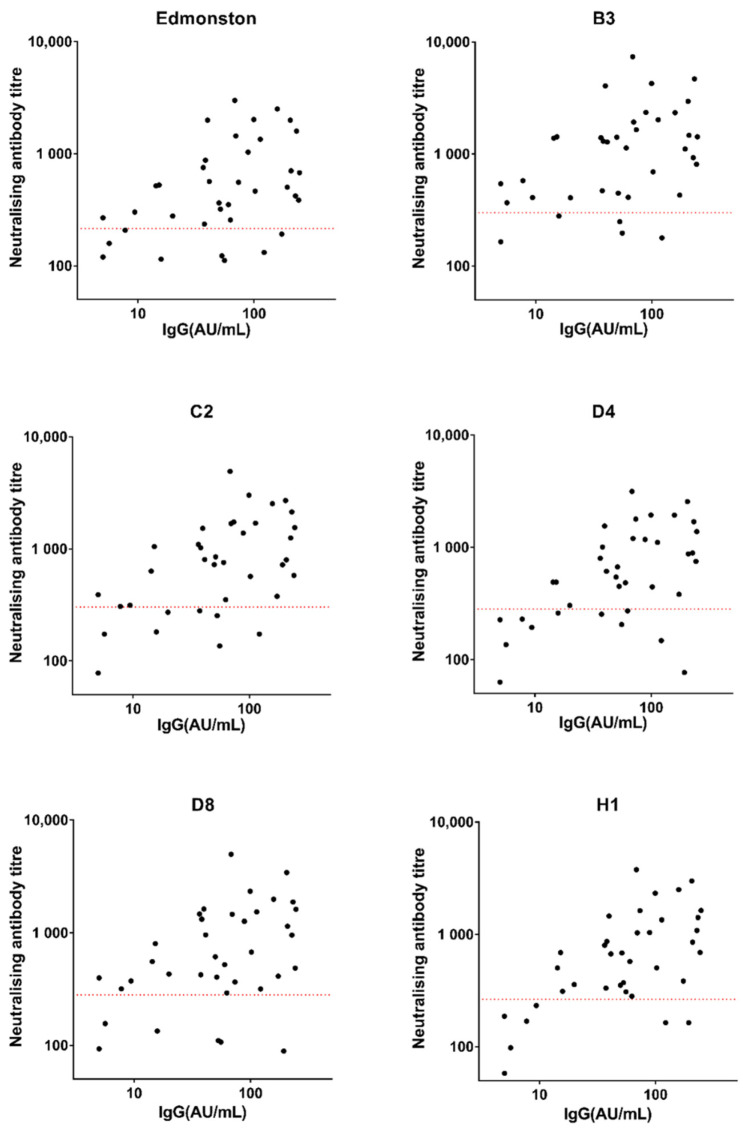
Measles-specific IgG levels (measured by kit from LIAISON^®^) and neutralising antibody titres in samples collected from Sheffield HCWs. The dotted red lines indicate the titre of the 3rd WHO reference serum (NIBSC 97/648) diluted to 120 mIU/mL, defined as the protective threshold.

**Figure 3 viruses-14-01716-f003:**
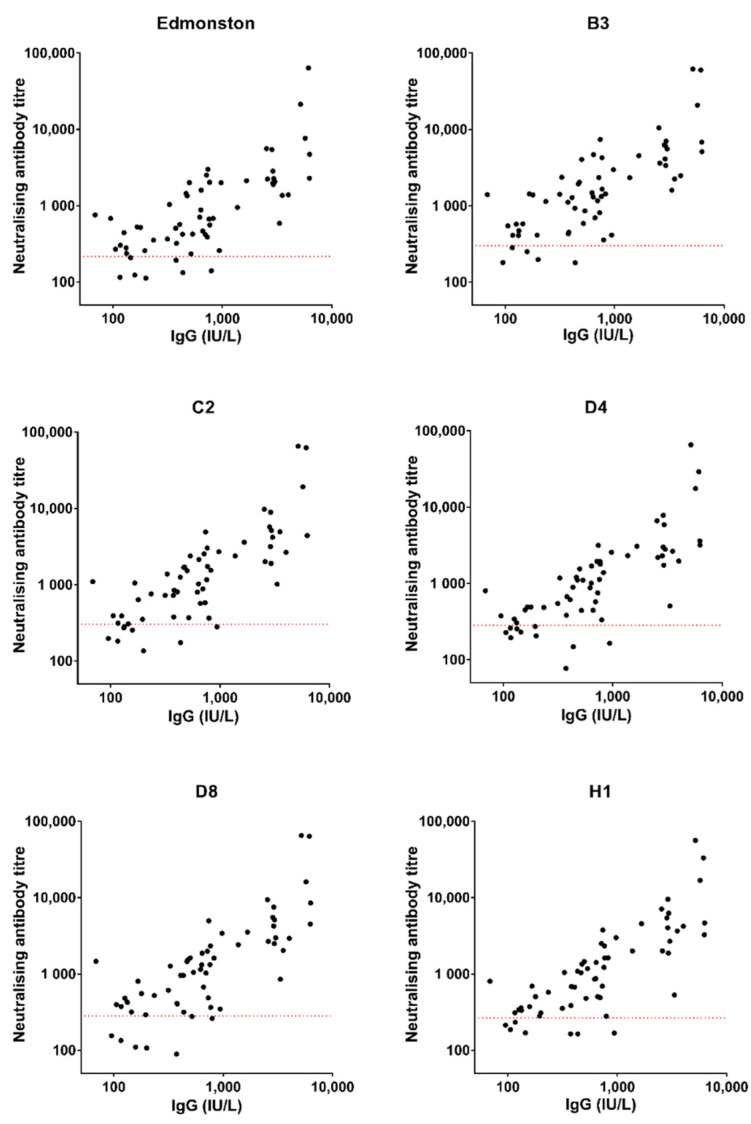
Measles-specific IgG levels (measured by kit from EUROIMMUN^®^) and neutralising antibody titres in Sheffield HCWs. The dotted red lines indicate the titre of the 3rd WHO reference serum (NIBSC 97/648) diluted to 120 mIU/mL, defined as the protective threshold.

**Table 1 viruses-14-01716-t001:** Proportion of Sheffield HCWs with neutralising antibody titres greater than the protective threshold against the MeV genotypes tested.

Genotypes	Percentage Protected (95%CI)
**Edmonston**	87.3 (76.6–93.7)
**B3**	93.6 (84.3–98.0)
**C2**	87.3 (76.6–93.7)
**D4**	82.5 (71.2–90.1)
**D8**	88.9 (78.5–94.8)
**H1**	88.9 (78.5–94.8)

**Table 2 viruses-14-01716-t002:** Spearman’s rank correlation coefficient between IgG levels (measured by kit from LIAISON^®^) and neutralising antibody titres in Sheffield HCWs.

Genotypes	R_s_	95%CI	*p* Value
**Edmonston**	0.42	0.098–0.66	0.0101
**B3**	0.41	0.094–0.66	0.0109
**C2**	0.49	0.19–0.71	0.0021
**D4**	0.51	0.21–0.72	0.0013
**D8**	0.40	0.075–0.65	0.0146
**H1**	0.55	0.26–0.74	0.0005

**Table 3 viruses-14-01716-t003:** Spearman’s rank correlation coefficient between IgG levels (measured by kit from EUROIMMUN^®^) and neutralising antibody titres.

Genotypes	R_s_	95%CI	*p* Value
**Edmonston**	0.71	0.55–0.82	<0.0001
**B3**	0.75	0.61–0.84	<0.0001
**C2**	0.78	0.66–0.87	<0.0001
**D4**	0.79	0.67–0.87	<0.0001
**D8**	0.76	0.62–0.85	<0.0001
**H1**	0.79	0.66–0.87	<0.0001

## Data Availability

Not applicable.

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
