# Peer review of "Correlating IgG Levels with Neutralising Antibody Levels to Indicate Clinical Protection in Healthcare Workers at Risk during a Measles Outbreak"

_viruses, 2022, doi:10.3390/v14081716_

Round 1

Reviewer 1 Report

The manuscript entitled: “Correlating IgG level with neutralising antibody level to indicate clinical protection in healthcare workers at risk during the measles outbreak” by Hu et al, submitted to journal Viruses, manuscript number 1808296, has been reviewed. It presents the results of a study on the levels of neutralizing antibodies in healthcare workers and the risk for infection with measles using a newly developed pseudo-neutralization assay. The manuscript has been well written, and data presented in a logical manner.

Comments from the reviewer:

- Both the Liaison and Euroimmun are commercial assays have cut-offs well above the protective levels as described also in this manuscript. Authors should have calibrated these assays against the WHO International Standard to determine the level at which the cut-off was set and translate assay units to international units.

- As the pseudo-neutralization assay is a new assay, authors should have conducted a calibration of this assay comparing it with conventional plaque reduction neutralization assay, which can be considered the Gold Standard for determining measles antibody levels https://www.technet-21.org/en/chapter-9-laboratory-testing-for-determination-of-population-immune-status/9-1-detection-of-virus-specific-igg-and-protective-immunity and onwards.

- It is unclear if the HCW have been vaccinated before exposure to the circulating D8 virus or any other virus. Authors should provide details on vaccination status of the study subjects.

- Authors speculate on the risk of transmission of measles breakthrough cases, and fail to provide data to confirm in the form of virus actually isolated from breakthrough cases.

- Authors speculate on reexposure to B3 virus increasing population immunity but actual data is lacking and speculation unfounded

- Authors do not discuss the role of cellular immunity in protection against measles, which may be particular relevant in persons with low or undetectable IgG antibodies. As stated in the reference above, measuring IgG levels is a proxy for real protection. Cellular immunity does play a role, particularly when antibody levels have waned below detectable levels.

Reviewer 2 Report

In the present study, Dr Hu and colleagues evaluated anti-measles IgG and neutralizing antibodies in serum collected from HCW during a measles outbreak in 2016.

Neutralizing antibodies titres were determined using VSV based pseudotype VN assay. Anti-measles IgG titers were determined using two commercially available kits.

Major comments

-The work presented here is similar to a previous study (PMID 26209410, J Clin Virol 2015), so there is lack of novelty, although in the previous study not the Euroimmun kits were used. In addition, comparison of neutralizing titers against different measles strains has also been performed previously.

-The results of the pseudotyped virus neutralization were not compared with the golden standard assay for measles neutralizing antibodies (PRN). This could be of interest for especially the samples with the low antibody titers.

-Line 230. Can the authors exclude that observed differences in neutralizing antibody titers between strains are due to intrinsic differences between viruses or the assays used? Ideally, unvaccinated, B3 infected (and other genotypes) sera should also be tested in these assays.

Minor comments:

-Line 44.  A reference should be included

-line 47-53. Please update with recent data.

-line 312. Measles is in multiple countries not an endemic disease.
